# New Therapeutic Approaches to and Mechanisms of Ginsenoside Rg1 against Neurological Diseases

**DOI:** 10.3390/cells11162529

**Published:** 2022-08-16

**Authors:** Yang Sun, Yantao Yang, Shasha Liu, Songwei Yang, Chen Chen, Meiyu Lin, Qi Zeng, Junpeng Long, Jiao Yao, Fan Yi, Lei Meng, Qidi Ai, Naihong Chen

**Affiliations:** 1Hunan Engineering Technology Center of Standardization and Function of Chinese Herbal Decoction Pieces, College of Pharmacy, Hunan University of Chinese Medicine, Changsha 410208, China; 2Pharmacy Department, Xiangtan Central Hospital, Xiangtan 411100, China; 3Department of Pharmacy, The First Hospital of Lanzhou University, Lanzhou 730000, China; 4Key Laboratory of Cosmetic, China National Light Industry, Beijing Technology and Business University, Beijing 100048, China; 5State Key Laboratory of Bioactive Substances and Functions of Natural Medicines, Institute of Materia Medica & Neuroscience Center, Chinese Academy of Medical Sciences and Peking Union Medical College, Beijing 100050, China

**Keywords:** ginsenoside, Rg1, inflammation, oxidative stress

## Abstract

Neurological diseases, including Parkinson’s disease (PD), Alzheimer’s disease (AD), Huntington’s disease (HD), stroke, cerebral infarction, ischemia-reperfusion injury, depression and, stress, have high incidence and morbidity and often lead to disability. However, there is no particularly effective medication against them. Therefore, finding drugs with a suitable efficacy, low toxicity and manageable effects to improve the quality of life of patients is an urgent problem. Ginsenoside Rg1 (Rg1) is the main active component of ginseng and has a variety of pharmacological effects. In this review, we focused on the therapeutic potential of Rg1 for improving neurological diseases. We introduce the mechanisms of Ginsenoside Rg1 in neurological diseases, including apoptosis, neuroinflammation, the microRNA (miRNA) family, the mitogen-activated protein kinase (MAPK) family, oxidative stress, nuclear factor-κB (NF-κB), and learning and memory of Rg1 in neurological diseases. In addition, Rg1 can also improve neurological diseases through the interaction of different signal pathways. The purpose of this review is to explore more in-depth ideas for the clinical treatment of neurological diseases (including PD, AD, HD, stroke, cerebral infarction, ischemia–reperfusion injury, depression, and stress). Therefore, Rg1 is expected to become a new therapeutic method for the clinical treatment of neurological diseases.

## 1. Introduction

*Panax ginseng* Meyer is a traditional Chinese herbal medicine in China. Its root or stem has been used in drug treatment for hundreds of years in many Asian countries [1]. It was originally used as food and a source of energy. It was gradually learned that *P. ginseng* has many pharmacological effects, especially in neurological system diseases. There are many ingredients in *P. ginseng*, including ginsenosides, ginkgo ketones, polysaccharides, polypeptides, glycoconjugates and other compounds. The main bioactive component of *P. ginseng* is ginsenosides, and they are responsible for most of the activities of *P. ginseng* [2,3]. Ginsenosides are divided into the following categories: protopanaxadiol (PPD), which includes Rb1, Rb2, Rb3, Rc, Rd, and Rg3; protopanaxatriol (PPT), which includes Rg1, Re, Rf, and Rg2 [4]; and oleanane (ginsenoside Ro). Ginsenoside Rg1 is one of the most important pharmacological active components of *P. ginseng*, and has low toxicity and few side effects, as well as neuroprotective effects [5]. There are various neuroprotective mechanisms of Rg1, including enhancement of nerve growth factor, anti-inflammation, antioxidation, anti-apoptosis, inhibited excitotoxicity, excessive calcium ion influx into neurons, maintaining the level of cellular ATP, and maintaining the integrity of the neuron structure [4]. There are many neurological diseases, including Alzheimer’s disease (AD), Parkinson’s disease (PD), amyotrophic lateral sclerosis, multiple sclerosis and cognition and memory impairments [6,7]. The pathogenesis of neurological diseases is complex and diverse, but most of them are related to the factors such as mitochondrial dysfunction, oxidative stress [8], protein aggregation, inclusion body formation [9], neuroinflammation [10], obesity [11], and hypopituitarism [12].

## 2. Ginsenosides

Ginseng is a perennial herb belonging to the Acanthopanax family, and has been used for 4500 years. It is distributed in more than 30 countries, mainly in Asia, and especially in China [13]. Since 2000, the research on *P. ginseng* in the field of neuroscience and neurology has developed rapidly [14]. It has been used to treat various diseases, such as chronic diseases, fatigue, and functional or organ-weakened Qi-deficiency states [15]. *P. ginseng* was first seen in Shennong *Herbal Classics* and has been used for more than 5000 years. *P. ginseng* contains many chemical components. Ginsenoside is the main active ingredient of *P. ginseng* [16]. Modern pharmacological research showed that ginsenoside has various effects, including anti-inflammation [3], antioxidant [17], anti-apoptotic [18,19], inhibition of platelet aggregation [19] and so on. Ginsenoside has a variety of pharmacological effects, and thus may treat many diseases including cardiovascular diseases [20], given its anti-angiogenic [2], anti-diabetic [19], anti-cancer [21], and neuroprotective effects [18]. The general structure of ginsenosides is a hydrophobic tetracyclic steroid skeleton, which is connected with sugar molecules and responsible for the hydrophilicity of the molecule [2]. Most of the activities of *P. ginseng* are mainly derived from ginsenosides [22]. Ginsenosides are divided into three classes according to their chemical structure: protopanaxadiols, protopanaxatriols, and oleanane (ginsenoside Ro) [23]. Protopanaxadiols include Rb1, Rb2, Rb3, Rc, Rd, Rg3, Rh2, and Rh3. Protopanaxatriols include Re, Rf, Rg1, Rg2, and Rh1 [3,22] (Figure 1).

## 3. Ginsenoside Rg1

Ginsenoside Rg1 (molecular formula: C_42_H_72_O_14_) is a monomer of a tetracyclic triterpenoid derivative [24]. Due to its unique chemical structure, Rg1 has a variety of pharmacological properties. In one study, Rg1 inhibited autophagy and endoplasmic reticulum stress to protect cardiac function [25]. Furthermore, Rg1 regulated the NLRP-3 signaling pathway and improved blood sugar in diabetic rats [26]. Rg1 may be a target for the treatment of type 2 diabetes mellitus by reducing the inflammatory response [27]. Rg1 attenuated hepatic insulin resistance induced by high fat and high glucose through reducing the inflammatory response [28]. Rg1 improved liver fibrosis. It was hypothesized that Rg1 may be a mechanism of inhibiting the epithelial–mesenchymal transition and the formation of reactive oxygen species (ROS) [29]. Rg1 inhibited the activation of platelet as well as arterial thrombosis [30]. Rg1 reduced lung inflammation and damage by inhibiting endoplasmic reticulum stress and inflammation [31]. Additionally, Rg1 plays an increasingly important role in the treatment of atherosclerosis [32]. Most importantly, there is a growing body of research about the role of Rg1 in the nervous system and neurological diseases, such as PD [4,33], cerebral ischemia and reperfusion injury [1], antidepressant [34], and AD [35], as well as anti-aging [36].

## 4. Neurological Diseases

Traditional neurological diseases are divided into two main groups: chronic neurodegenerative diseases and acute neurodegenerative diseases [37]. The pathogenesis of neurodegenerative diseases, including PD [4], AD [35] and Huntington’s disease (HD) [38], is complex and relatively difficult to treat. The neurodegeneration of neurons is the gradual loss of structure and function, including neuronal death, leading to cognitive impairment [37]. Acute neurodegenerative diseases mainly include cerebrovascular accidents, including stroke [39], cerebral infarction [37], and ischemia-reperfusion injury [40]. Depression has become a common neurological condition, affecting one in five people worldwide [41,42]. Stress is another neurological disease that poses a serious threat to physical and mental health. Studies have confirmed that there is a close relationship between neuroinflammation and the activation of neuronal microglia. More importantly, there is growing evidence that stress is associated with neuronal microglia activation and neuroinflammation [43]. Chronic social frustration stress is strongly associated with depression. Fan et al. have shown that social stress leads to negative memory hippocampus [44]. The main molecular mechanisms in PD include α-synuclein misfolding and aggregation, mitochondrial dysfunction, neuronal injury protein clearance (associated with ubiquitin proteasome and autophagy lysosome system defects), neuroinflammation, as well as oxidative stress [45]. In the “gut brain” phenotype of PD, the disease can be identified as “gut syndrome”, in which a variety of factors, such as microbial disorders, intestinal leakage/endotoxemia and intestinal inflammation, can lead to peripheral α-synuclein disease and intestinal dysfunction. This shows that there is a close relationship between the gastroenterological and neurological manifestations of PD [46].

The main neuropathological features of AD include neurofibrillary tangles, amyloid plaques, dystrophic neurites, the accumulation of nerve fibers, and other deposits found in the brain of AD patients, along with large atrophy caused by the loss of nerves, nerve cells and synapses. Other factors can cause neurodegeneration, such as neuroinflammation, oxidative stress, and cholinergic neuron damage [47]. Animal models and clinical studies have proven that inflammation has an irreplaceable role in AD. Systemic inflammation in AD patients or animals may aggravate neurological deficits. In addition, inflammatory factors can also promote the permeability of the blood–brain barrier [48]. During the aggregation process, ROS is formed, which leads to the dysfunction of lipid peroxidation glucose transporter and ion channel ATPase in the neuronal cell membrane. This oxidative stress induces homeostasis and metabolism, making neurons vulnerable to apoptosis in AD [49].

The production of aggregates is the most significant indicator of HD. Neuronal and synaptic abnormalities occur in the early stages of HD disease. HD aggregates were first found in the nucleus, and later, with the advancement of research, they were also found to occur in the cytoplasm and neuronal processes in the brain of HD patients. Neurodegeneration is related to mitochondrial dysfunction, which leads to the production of ROS, which causes more damage to mitochondria in HD. Huntington protein is expressed in immune cells, leading to cell autonomous microglia activation and proinflammatory cytokine secretion. In addition, in the peripheral immune system, mutant Huntington protein acts on the NF-κB signaling pathway and affects the occurrence of inflammatory response [50]. In addition, excessive extracellular glutamate will lead to continuous stimulation of N-methyl D-aspartic acid (NMDA) receptors and neuronal death, which will also induce HD [51].

Stroke is the second leading cause of death in the world. Stroke is divided into two categories. The first is ischemic stroke, accounting for about 85% of stroke cases, and the second is hemorrhagic stroke. The causes of ischemic stroke include inflammation, energy failure, loss of homeostasis, acidosis, elevated intracellular calcium levels, excitotoxicity, free radical-mediated toxicity, cytokine-mediated cytotoxicity, complement activation, damage to the blood–brain barrier, activation of glial cells, oxidative stress and leukocyte infiltration. Hemorrhagic stroke has a high mortality rate. The main causes are hypertension, vascular system damage, and excessive use of anticoagulants and thrombolytic agents [52]. Inflammation caused by ischemia lasts longer in the brain. This chronic inflammation is neurotoxic and may cause damage to peripheral neurons. In addition, stroke can cause fatigue, depression and dementia [53]. The time period after ischemic stroke, the severity of ischemia, systemic blood pressure, the venous system, and the location of the infarction will affect the infarction area and nervous system severity after ischemic stroke. Ischemia and reperfusion lead to the activation of cell death programs, including apoptosis, autophagy, and necrosis. Ischemia is associated with the activation of hypoxia and inflammatory signals, including hypoxia inducible factor (HIF), the NF-κB signaling pathway, mitogen-activated protein kinase (MAPK), and type I interferon pathways, thereby inducing the activation of proinflammatory cytokines and chemokines during ischemia-reperfusion. Oxidative stress can activate TLR4, which is produced by ischemia and reperfusion, and can trigger inflammatory cells to produce inflammatory reactions [54]. Depression is a kind of persistent mental disease. Studies have shown that depression is related to the expression of a variety of molecules, including IL-6, IL-2, IL-1β, TNF-α, and C-reactive protein (CRP) [55]. The decrease in brain-derived neurotrophic factor (BDNF) and insulin-like growth factor (IFG-I) will increase the risk of depression. In addition, the sensitivity of the 5-hydroxytryptamine (5-HT) receptor is reduced in patients with depression. Increased production of nitric oxide, mitochondrial dysfunction, and oxidative stress occur in the disease process of patients with depression [56]. Studies have shown that the production of stress is related to the activation of microglia and the occurrence of neuroinflammation [43].

There are many kinds of neurological diseases, and their pathogenesis is complex. Nevertheless, the pathogenesis of most neurological diseases is closely related to apoptosis, neuroinflammation, oxidative stress, and the MAPK family. Therefore, by intervening in these molecules and signal transductions, we could advance the treatment of neurological diseases.

## 5. Mechanisms

### 5.1. Effects of Rg1 on Apoptosis

Studies have demonstrated that Rg1 pretreatment inhibited the increase in the expression levels of caspase-3 and caspase-9, as well as the decrease in the expression level of B-cell lymphoma-2 (Bcl-2) caused by chronic unpredictable mild stress (CUMS). In addition, Rg1 inhibited the expression of nuclear factor-erythroid 2-related factor 2 (Nrf2) and the activation of p38 mitogen-activated protein kinase (p-p38 MAPK) and the nuclear factor-κB (NF-κB/p65) subunit in CUMS [44]. The combined use of Rg1 and acori graminei Rhizoma inhibited neuronal apoptosis in Alzheimer’s mice by regulating the expression of miR-873-5p [57]. In previous studies, chemically induced aging mice were established by the consecutive administration of D-galactose and AlCl_3_. Rg1 restored the fibroblast growth factor 2 (FGF2)-Protein Kinase B (Akt) and the brain-derived neurotrophic factor (BDNF)—Tyrosine Kinase receptor B (TrkB) signaling pathways in the hippocampus and prefrontal cortex to inhibit neuronal apoptosis [36]. Studies have shown that lidocaine induced transient neurological symptoms and cauda equina syndrome. The results suggest that the protective effect of Rg1 on lidocaine-induced apoptosis is mediated by downregulating caspase-3 expression [58]. In addition, it has been confirmed that Rg1 inhibits apoptosis induced by amyloid-beta peptide fragment 25–35 (Aβ25–35) through Akt and extracellular-regulated protein kinases (ERK) signaling [59]. Rg1 reduced the production of dopamine-induced ROS and the release of mitochondrial cytochrome-C into the cytoplasm, and subsequently inhibited the activation of caspase-3. Rg1 resisted apoptosis of pheochromocytoma 12 (PC12) cells induced by exogenous dopamine, which provided a new target and approach for the treatment of PD [60]. Rg1 inhibited hippocampal neuronal apoptosis in depressed mice by promoting G-protein-coupled estrogen receptor 1 (GPER) [61] (Figure 2). In general, many studies have shown that Rg1 can improve neurological diseases by inhibiting apoptosis. However, the inhibition of the type of apoptosis and the specific site of action are not clear. The existing research shows that neurons represent the cell type that is more involved. In addition, Rg1 can inhibit apoptosis in neurodegenerative diseases, and it plays a preventive role in affective diseases such as stress.

### 5.2. Effects of Rg1 on Neuroinflammation

Neuroinflammation is mainly associated with the presence of activated microglia in the brain and increased levels of central nervous system cytokines. Peripheral inflammation could trigger microglia to enter a pro-inflammatory phenotype, leading to a stronger central nervous system response [62]. Therefore, it is essential to reduce the neuroinflammatory response in neurological diseases. Studies have demonstrated that Rg1 ameliorated chronic social defeat stress-induced (CSDS) depressive-like behaviors. The molecular mechanism was that Rg1 inhibited the release of pro-inflammatory cytokines interleukin (IL)-6, IL-1β, and tumor necrosis factor-α (TNF-α), and inhibited NF-κB expression via the mitogen-activated protein kinase (MAPK) and NAD-dependent deacetylase sirtuin-1 (SIRT1) signaling pathways. Rg1 treatment also inhibited the activation of microglia in the hippocampus, suggesting that Rg1 reduces the neuroinflammatory response via MAPK and SIRT1 signaling pathways and results in the inhibition of NF-κB transcriptional activity in CSDS [63]. Lipopolysaccharide (LPS)-induced systemic inflammation is another relevant area of research. Animal models of LPS were used to study neuroinflammation and PD. Rg1 ameliorated neuroinflammation by regulating microglia polarization dynamics and nuclear translocation of NF-κB in LPS-induced PD. Rg1 reduced pro-inflammatory cytokines and increased anti-inflammatory cytokines including transforming growth factor (TGF-β) and IL-10 as well as neurotrophic factor (BDNF) secretion to protect neurons in PD. However, the specific protective mechanism needs further study [64]. Neuroinflammatory disorders have a close relationship with depression. In particular, studies have confirmed that Rg1 inhibited the release of inflammatory factors, reduced the neuroinflammatory response, and combated depression. Rg1 reduced the level of intracellular adhesion molecule 1 (ICAM-1), cyclooxygenase-2 (COX-2), and inducible nitric oxide synthase (iNOS), as well as maintained the integrity of BBB permeability in neuroinflammation-induced behavioral deficits [65]. Studies have shown that Rg1 reduced the levels of IL-1β, TNF-α, caspase-1, IL-2, IL-6 and IL-18 via the suppression of Connexin43 (Cx43) ubiquitination in depression [66] (Figure 3). Overall, there are few disease model studies on Rg1 inhibiting the neuroinflammatory response, and it is more about inhibiting the expression of inflammatory factors.

### 5.3. Effects of Rg1 on microRNA (miRNA) Family

MiRNA is a small set of endogenous non-coding RNAs. In one study, Rg1 protected ischemic/reperfusion-induced neuronal injury by regulating miR-144, which regulated the Nrf2/antioxidant response element (ARE) signal via miR-144 [67]. Some studies have suggested that the neuroprotective effects of Rg1 were caused by the regulation of the miR-134 signaling pathway in chronic stress-induced structural plasticity and depression-like behaviors. It blocked the function of miR-134 and significantly improved neuronal structural abnormalities, biochemical changes, and depression-like behavior [68]. In addition, Rg1 inhibited apoptosis in AD by regulating miR-873-5p [57]. There are few studies on Rg1 improving neurological diseases through the miRNA family. In addition, the members of the miRNA family involved only include miR-134, miR-144, and miR-873-5p. Interestingly, recent studies have shown that Rg1 can reduce the destruction of BBB and traumatic brain injury by inhibiting the production of exosomal miR-21 [68]. This can lead us to further study the mechanism of Rg1 on the miRNA family in neurological diseases in the future (Figure 4).

### 5.4. Effects of Rg1 on Mitogen-Activated Protein Kinase (MAPK) Family

The MAPK family includes ERK1/2, c-Jun N-terminal kinase (JNK), and p38 MAPK, which are involved in various physiological and pathological processes [63]. Rg1 plays a neuroprotective role by activating the Nrf2 signaling pathway and inhibiting the activation of NF-κB and p38 MAPK, thereby reducing neuroinflammation and apoptosis in CUMS-induced depressed rats [49]. Rg1 induced the growth of cultured hippocampal neurons through the ERK signaling pathway. In addition, Rg1 prevented Aβ25–35-induced apoptosis via ERK signaling [59]. Moreover, Rg1 significantly inhibited the levels of ERK1/2, JNK, and p38 MAPK, and inhibited the release of proinflammatory cytokines in chronic social defeat stress (CSDS) [63]. Rg1 inhibited the expression of high-voltage-activated calcium currents (I_Ca,HVA_) in hippocampal neurons of Aβ-exposed rat brain slices through MAPK [69]. Rg1 inhibited proinflammatory monocytes, which inhibited C-C motif chemokine-2 (CCL2)-induced activation of p38/MAPK and phosphatidylinositol 3-kinase (PI3K)/protein kinase B (Akt) pathways. The anti-depression mechanism of Rg1 can be illustrated by the attenuation of Ly6Chi monocyte pro-inflammatory factor release in the blood [70] (Figure 5). Rg1 improves neurological diseases by inhibiting the expression of MAPK family molecules and more by inhibiting the neuroinflammatory response. Rg1 seems to be closely related to inhibiting neuritis and improving neurological diseases, especially neurodegenerative diseases.

### 5.5. Effects of Rg1 on Oxidative Stress

Ye et al. measured changes in oxidative stress levels in depressed rats induced by CUMS. They found that Rg1 enhanced the activity of antioxidant enzymes such as superoxide dismutase (SOD) and glutathione peroxidase (GSH-pX) while reducing the level of ROS and malondialdehyde (MDA) in the hippocampal CA1 region of depressed rats. These results suggest that Rg1 attenuated oxidative stress in the hippocampal CA1 region of depressed rats [71]. NADPH oxidase 2 (NOX2) is a major source in the neurological system. Rg1 decreased ROS production, and reduced the level of NOX2 in H_2_O_2_-treated hippocampal neurons [72]. Rg1 ameliorated dopamine-induced apoptosis in PC12 cells by inhibiting oxidative stress [17]. Rg1 downregulated the oxidative stress marker 4-hydroxytryptamine (4-HNE) in the CA1 region of the hippocampus in depressed rats [71]. Rg1 reduced the level of ROS and increased the activities of SOD and GSH-pX in vivo and in vitro, so as to improve the cognitive impairment of mice and delay the aging of neural stem cells [72]. In addition, studies have confirmed that this protected neurons by reducing oxidative stress [73].

Recent research showed that Rg1 alleviates cadmium-induced neurotoxicity in vitro and in vivo by attenuating oxidative stress and inflammation [74]. Rg1 improved oxidative stress, apoptosis, and neuroinflammation in AD [75] (Figure 6). Therefore, Rg1 can inhibit oxidative stress by acting on neuronal cells in vitro. Rg1 can inhibit oxidative stress and alleviate depression-like behavior by acting in the hippocampal CA1 region in vivo. This improvement is closely related to the inhibition of apoptosis.

### 5.6. Effects of Rg1 on NF-κB

NF-κB is a transcription factor that causes the polarization of microglia M1 [76]. There is increasing evidence showing that NF-κB is closely related to the regulation of the inflammatory response. After activation, NF-κB enters the nucleus, binds to the IκB sequence in the promoter region of various cellular inflammatory factors, and participates in NO synthases and the transcription of inflammatory mediators (IL-2, IL-6 and TNF-α) [77]. Microglia-mediated neuroinflammation is one of the most significant features of PD animal models [78]. NF-κB is also considered to be the central transcription factor of inflammatory mediators. Moreover, it plays an important role in the activation of microglia [79]. Some studies have found that Rg1 restrained LPS-induced NF-κB activation in PD mice, and that Rg1 mediated the differentiation of microglia into the M2 phenotype via the NF-κB pathway [64]. Previous studies have confirmed that the accumulation of amyloid beta (Aβ) is an early event in the development of AD [80]. It was demonstrated that Rg1 had a neuroprotective effect on primary cultured rat cortical neurons cells resisting Aβ25–35 damage by interfering with mitochondria apoptotic pathways downregulated via the NF-κB/NO signal pathway [81]. HD is one of the neurodegenerative diseases characterized by striatal atrophy, involuntary movement and uncoordinated movement [82,83]. It was found that Rg1 prevented the death of striatal neurons by inhibiting the activation of MAPKs and the NF-κB pathway in HD, which suggested that Rg1 had a neuroprotective effect in HD mouse models through these mechanisms [84]. In general, NF-κB is closely related to neurodegenerative diseases, including AD, PD, and HD. Rg1 inhibits NF-κB expression, and thus improves neurodegenerative diseases. Rg1 may eventually inhibit the neuroinflammatory response and play a protective role.

### 5.7. Effects of Rg1 on Learning and Memory

Synaptosomal-associated protein 25 (SNP25) is a pivotal marker of synaptic function and one of the main proteins formed by the neurosoluble N-ethylmaleimide sensitive factor attachment protein receptor (SNARE) complex [85]. The SNARE complex plays a vital role in the central nervous system, and it is also crucial in the formation of learning and memory [86]. It has been found that the expression of SNP25 in the hippocampus of triple-transgenic mice of AD (3 × Tg-AD) was significantly lower than that of wild-type mice. Rg1 regulated the protein expression in 3 × TG AD mice and improved their memory impairment, suggesting a protective effect of Rg1 on memory [87]. Senescence-accelerated mice P8 (SAMP8) is a classic accelerated aging mouse model [88], which shows age-related memory and learning functions [89]. Rg1 ameliorated the escape acquisition and reversed memory deficits of SAMP8 mice [90]. In addition, Rg1 also improved learning and memory in chronic restraint stressed (CRS) rats [91] (Figure 7). Therefore, we can conclude that Rg1 improves learning and memory function in AD and geriatric disease models. However, its specific mechanisms and potential roles remain unclear.

## 6. Targets of Neurological Diseases

Neuroinflammation in the central nervous system (CNS) is an important topic in the study of neuroimmunology [92]. Inflammatory diseases of the peripheral nervous system (PNS) and CNS are very common, and greatly affect the physical and mental health of the affected people [93]. Inflammation can induce mitochondrial dysfunction and, conversely, inflammation can be induced by mitochondrial dysfunction. Mitochondrial dysfunction and neuroinflammation may contribute to the development of neurodegeneration and neurodegenerative diseases [94]. In recent years, numerous evidence has confirmed that the upregulation of autophagy may have a protective effect on neurodegeneration. However, autophagy dysfunction is also involved in the pathogenesis of many diseases [95]. The microbiota of the large intestine constitute a complex micro-ecosystem in the host, and microbiota stability is closely related to human health [96]. Changes in microbial composition may change signals in the central nervous system. In one study, Rg1 improved cognitive impairment by increasing the number of Lactobacillus salivarius in AD [97]. It has been shown that Rg1 changed the abundance of intestinal flora to improve AD, and that proteobacteria and verrucomicrobia are key microbiota [98] (Figure 7). Vascular endothelial growth factor (VEGF) is an effective growth factor, which plays a variety of roles in angiogenesis. VEGF has neuroprotective and pathological effects in ischemic and hemorrhagic stroke [99]. Caveolin-1 (Cav-1) is an integrin located on the sponge membrane. It has been proven that Cav-1 protected the integrity of the BBB by inhibiting matrix metalloproteinase (MMP), which degrades tight junction proteins. Cav-1 reduced the permeability of the BBB by downregulating MMP9, reduced neuroinflammation by affecting cytokines and inflammatory cells, promoted nerve regeneration and angiogenesis through the Cav-1/VEGF pathway, and reduced apoptosis and oxidative stress-mediated damage [100]. PPARs (peroxisome proliferator-activated receptors) are ligand-activated transcription factors which play an important role in cellular processes such as cell differentiation and metabolism. PPAR activation has demonstrated a beneficial role in many preclinical models of neurodegenerative diseases and central nervous system injury [101]. There is increasing evidence that insulin is associated with several physiological functions of the brain, such as food intake and weight control, reproduction, learning and memory, neuromodulation, as well as neuroprotection. In addition, it is now clear that insulin and insulin disorders may play a role in the development and progression of neurodegenerative diseases and neuropsychiatric diseases [102].

In conclusion, the pathogenesis of nervous system diseases is complex. Therefore, it is essential to search for drugs with definite curative effects, less toxicity and fewer side effects. Rg1 is the main component of *P. ginseng*. It has been proven that Rg1 regulates a variety of cell signaling pathways and has pharmacological effects. In one study, Rg1 administration alleviated Adriamycin-induced heart failure by inhibiting excessive autophagy and endoplasmic reticulum stress [103]. Rg1 protected mice from STZ-induced type 1 diabetes by regulating NLRP3 [26]. Rg1 inhibited post-traumatic stress disorder (PTSD)-like behavior in mice by promoting synaptic protein [104]. Rg1 reduced hepatic insulin resistance induced by high fat and high glucose by inhibiting inflammation [28]. Rg1 reduced sepsis-induced lung inflammation and damage by inhibiting endoplasmic reticulum stress and inflammation [31]. Rg1 improved liver fibrosis by inhibiting epithelial and mesenchymal transition (EMT) and production of ROS in vivo and in vitro [32]. Rg1 has neuroprotective effects and low toxicity. It also has beneficial effects on many neurological diseases [4]. In AD animal models, Rg1 reduced the level of brain Aβ, reduced oxidative stress, enhanced free radical scavenging, inhibited apoptosis caused by Aβ accumulation, and kept the neuronal activity and plasticity of the hippocampus in a normal state [35]. Studies have shown that Rg1 inhibited morphine-induced free movement and spatial memory impairment in anesthetized rats, and upregulated mouse hippocampal neurogenesis to produce antidepressant effects by activating the BDNF signaling pathway [87]. Rg1 has many pharmacological effects and can act on a variety of signal pathways. As the pathogenesis of nervous system-related diseases is complex and affected by a variety of signal pathways, the future research on Rg1 can start with the pathogenesis related to the nervous system, find other targets of Rg1, and then facilitate in-depth studies of specific pathways and targets (Table 1).

## 7. Discussion

### 7.1. Possible Crosstalk and Perspective

The pathogenesis of nervous system-related diseases is complex, and different signal pathways interact with each other. Studies have confirmed that Rg1 can treat AD by regulating the metabolism of intestinal flora [97,98]. However, there are few studies on Rg1 improving neurological diseases through intestinal flora. This is a direction of future research and worthy of our attention.

### 7.2. In-Depth Study of Rg1

Given that the complexity of the chemical structure of ginsenoside is complex, can we further improve its therapeutic effect by changing its structure–activity relationship? Can we also improve the efficacy and bioavailability by changing the dosage form or making compound preparations with other drugs? In addition, Rg1 has a variety of pharmacological effects, but the research on ginsenosides is controversial. Whether ginsenosides are soluble in water is a matter of great concern. In future research on Rg1, it would be valuable to focus on whether it is soluble in water, so as to improve the bioavailability. Ginsenosides have many chemical components [3,23]. Rg1 can be biotransformed through deglycosylation in the intestine, and the metabolites of Rg1 have greater biological effects than Rg1. However, pharmacokinetic studies have shown that the oral bioavailability of Rg1 is very poor. The amount of Ginsenoside Rg1 absorbed by oral administration ranges from 1.9% to 20.0% of the dose. The poor oral bioavailability of Rg1 and its metabolites may be due to poor membrane permeability and active bile excretion [106]. Research showed that Rg1 can penetrate the BBB, but it was not very skilled in doing so [107]. In addition, research confirmed that Rg1 nanoparticles penetrate the BBB [108]. Other studies have confirmed that Rg1 improved neuroinflammation. In the future, we can study whether other components in ginsenoside also play a neuroprotective role, if Rg1 should be used alone or in combination with other treatments, which effects will be significant, and if the toxicity and side effects of Rg1 will be minor. Finding drugs with significant curative effects, low toxic, and few effects is an urgent problem to be solved in the treatment of nervous system-related diseases. There are few studies on the pharmacokinetics and toxicity of Rg1, so further research is needed. Moreover, in recent years, research on the nervous system has become a hotspot, especially anti-neuroinflammatory mechanisms. In-depth research on Rg1 can provide new targets and approaches for the clinical treatment of neurological diseases, which opens up new horizons and ideas for the development of ginsenosides and Rg1.

### 7.3. The Effect of Rg1 in Neurological Diseases

Rg1 has been shown to improve neurological diseases through a variety of pharmacological effects, including apoptosis, miRNA, neuroinflammation, oxidative stress, the MAPK family, NF-κB, and learning and memory. Rg1 involves many effects and mechanisms in neurological diseases. Research has confirmed that Rg1 inhibited neuronal apoptosis in AD [57]. In addition, Rg1 induced neuronal apoptosis in the hippocampus and cortex of aged rats [36] and inhibited hippocampal neuronal apoptosis in depressed mice [63]. This draws us to other parts of the brain, such as the striatum, or other cell types (microglia and morphoglia)—can we continue to further study the effect mechanisms of Rg1 on different parts of the brain and different regions in neurological diseases, and further study the effect mechanisms of Rg1 on specific cell types in neurological diseases?

Research has confirmed that Rg1 can inhibit neuroinflammation in neurological diseases [63,65]. However, most studies focus on the mechanism of pro-inflammatory factors [63,65,66], and few studies examine anti-inflammatory factors [58]. In future research, we can further study the effect of Rg1 on anti-inflammatory factors in neurological diseases, and how Rg1 affects the balance between anti-inflammatory and proinflammatory factors to improve neurological diseases.

Rg1 improves neurological diseases by affecting miRNA, including miR-134, miR-144, and miR-873-5p [57,67,68]. It seems that Rg1 has a close relationship with these three miRNAs in neurological diseases. Thus, we suggest an in-depth study on the improvement effect of Rg1 on neurological diseases among these three miRNAs (miR-134, miR-144, and miR-873-5p) or other miRNAs—whether Rg1 could affect the regulation of different target genes by the same miRNA in neurological diseases, and whether Rg1 can affect the regulation of the same target gene by different miRNAs.

The MAPK family includes ERK1/2, JNK, and p38 MAPK, which are involved in neurological disease [63]. More importantly, the MAPK family seems to be closely related to inflammation and apoptosis [44,59].

Some articles have shown that the interaction between oxidative stress and neuroinflammation contributes to the development of dopamine neurodegeneration [109]. Oxidative stress-mediated neuronal apoptosis promotes cognitive impairment in rats [110]. Can Rg1 inhibit crosstalk between different signaling pathways to improve neurological diseases? It would be valuable to further study whether Rg1 could inhibit oxidative stress and then inhibit inflammation or apoptosis in nervous system-related diseases.

Research has shown Rg1 to inhibit the NF-κB signaling pathway and that it improved neurological diseases (AD, PD, and HD) 64,80,85,86]. Rg1 has also improved learning and memory functions in neurodegenerative disease models. However, the specific mechanisms and potential roles remain unclear.

## 8. Conclusions

There are many kinds of neurological diseases, and their pathogenesis is complex. Such diseases have high incidence and mortality rates, thus endangering the physical and mental health of patients. Therefore, it is imperative to find safe and effective drugs with low toxicity and manageable side effects. An increasing amount of evidence shows that Rg1 plays an important role in neurological diseases. Rg1 acts on neurological diseases through a variety of signal pathways and related molecular mechanisms in conditions such as PD, AD, HD, stroke, cerebral infarction, ischemia-reperfusion injury, depression, and stress. Rg1 is thus expected to become a drug for the treatment of neurological diseases, and may become a new therapeutic strategy and means for the clinical treatment of neurological diseases.

## Figures and Tables

**Figure 1 cells-11-02529-f001:**
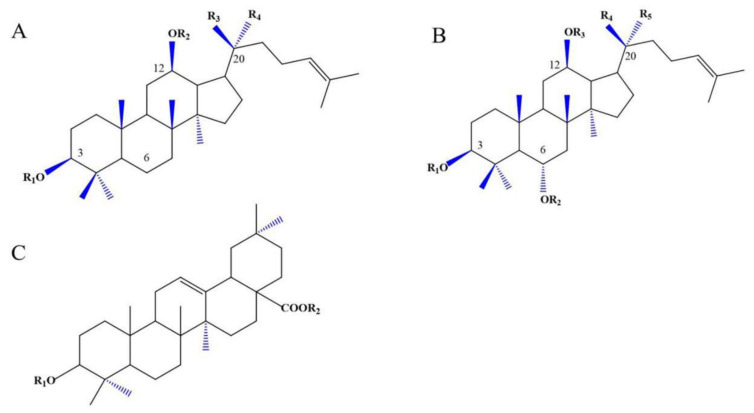
The three basic structures of Ginsenoside. (**A**) Protopanaxadiols, (**B**) protopanaxatriols, (**C**) oleanane.

**Figure 2 cells-11-02529-f002:**
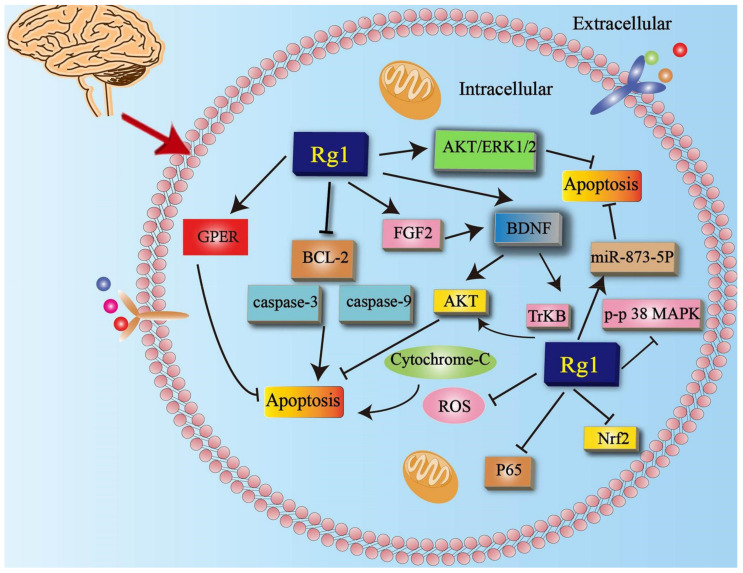
Rg1 improves nervous system diseases by inhibiting apoptosis. Rg1 inhibits the expression levels of caspase-3 and caspase-9, as well as the decrease in the expression level of Bcl-2. Rg1 inhibits apoptosis by upregulating the level of BDNF, GPER, FGF2, and AKT/ERK1/2 while increasing the level of miR-873-5p. Akt = protein kinase B;ERK1/2 = extracellular-regulated protein kinases 1/2; BDNF = brain-derived neurotrophic factor; GPER = G-protein-coupled estrogen receptor 1; TrkB = Tyrosine Kinase receptor B; ERK = extracellular-regulated protein kinases; FGF2 = fibroblast growth factor 2; Bcl-2 = B-cell lymphoma-2; ROS = reactive oxygen species; Nrf2 = nuclear factor-erythroid 2-related factor 2; p-p38 MAPK = p38 mitogen-activated protein kinase; p65 = nuclear factor-κB.

**Figure 3 cells-11-02529-f003:**
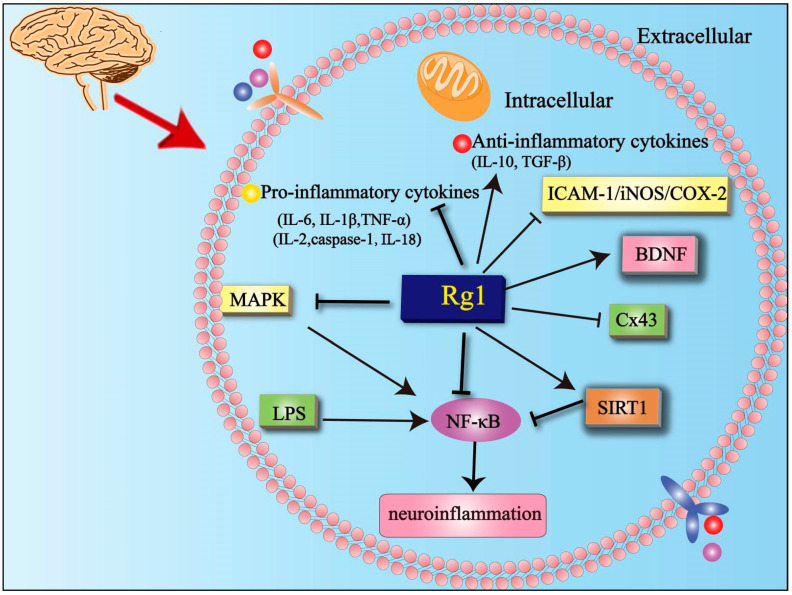
Anti-neuroinflammatory mechanism of Rg1 in neurological diseases. Rg1 reduces neuroinflammation in neurological diseases by inhibiting the level of NF-κB and MAPK while upregulating the expression of SIRT1. NF-κB = nuclear factor-κB; SIRT1 = sirtuin-1; LPS = Lipopolysaccharide; TNF-α = tumor necrosis factor-α; IL = interleukin; BDNF = neurotrophic factor; Cx43 = Connexin43; ICAM-1 = intracellular adhesion molecule 1; COX2 = cyclooxygenase-2; iNOS = inducible nitric oxide synthase.

**Figure 4 cells-11-02529-f004:**
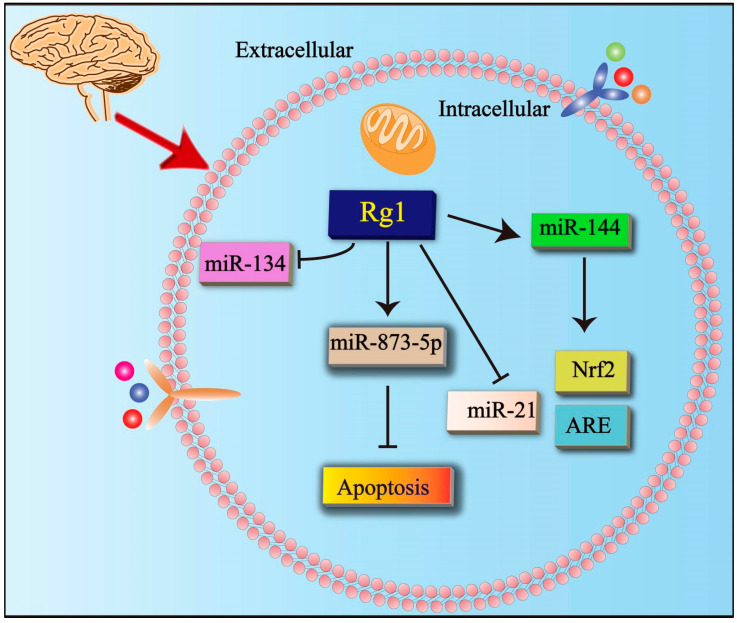
Effects of Rg1 on microRNA (miRNA) family. Rg1 inhibits the expression of miR-134 while upregulating the expression of miR-144 and miR-873-5p in neurological diseases. Rg1 inhibited apoptosis through miR-873-5p. Rg1 upregulates ARE and Nrf2 through miR-144. Rg1 can reduce the destruction of BBB and traumatic brain injury by inhibiting the production of exosomal miR-21. Nrf2 = nuclear factor-erythroid 2-related factor 2; ARE = antioxidant response element; miR-144 = microRNA 144; miR-873-5p = microRNA-873-5p; miR-134 = microRNA-134; miR-21 = microRNA-21.

**Figure 5 cells-11-02529-f005:**
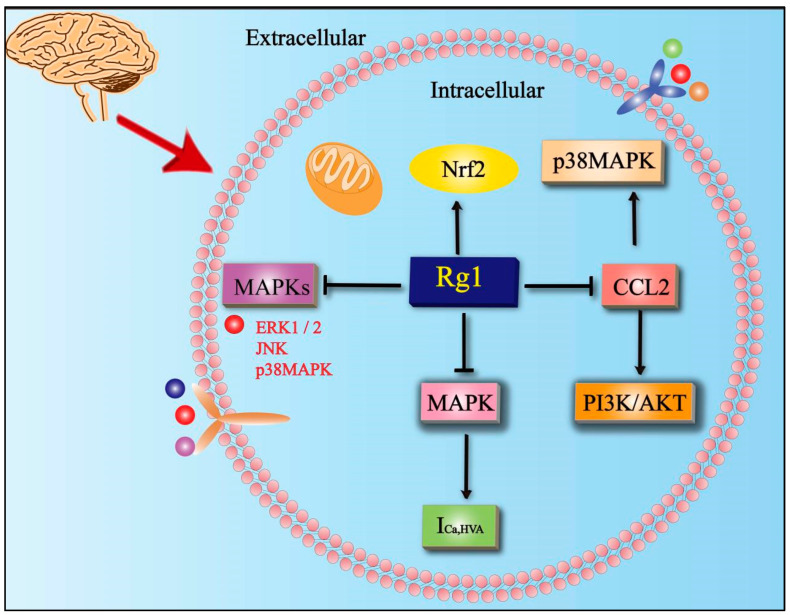
Rg1 inhibits the expression of CCL-2 and the MAPK (ERK1/2, JNK, and p38 MAPK) family in neurological diseases. MAPK = mitogen-activated protein kinase; CCL2 = C-C motif chemokine-2; I_Ca,HVA_ = high-voltage-activated calcium currents; ERK1/2 = extracellular-regulated protein kinases 1/2; JNK = c-Jun N-terminal kinase; PI3K = phosphatidylinositol 3-kinase; Akt = protein kinase B; Nrf2 = nuclear factor-erythroid 2-related factor 2.

**Figure 6 cells-11-02529-f006:**
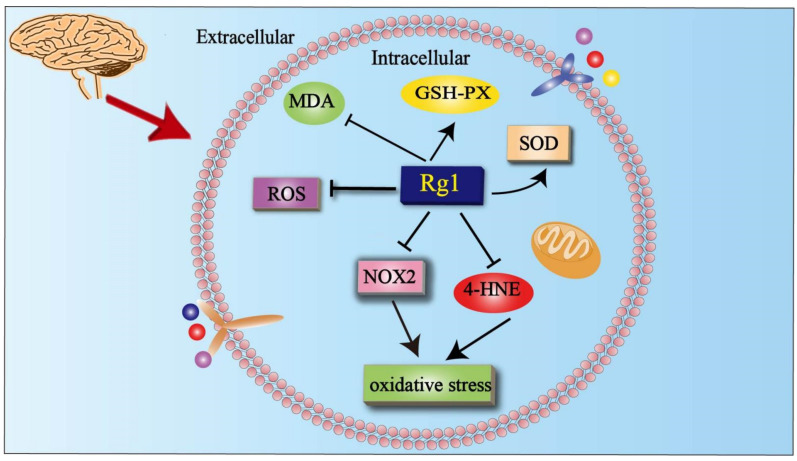
Effects of Rg1 on oxidative stress. Rg1 improves oxidative stress in neurological diseases by reducing the level of MDA, ROS, NOX2, and 4-HNE while increasing the level of SOD and GSH-px. SOD = superoxide dismutase; GSH-pX = glutathione peroxidase; MDA = malondialdehyde; NOX2 = NADPH oxidase 2; 4-HNE = 4-hydroxytryptamine; ROS = reactive oxygen species.

**Figure 7 cells-11-02529-f007:**
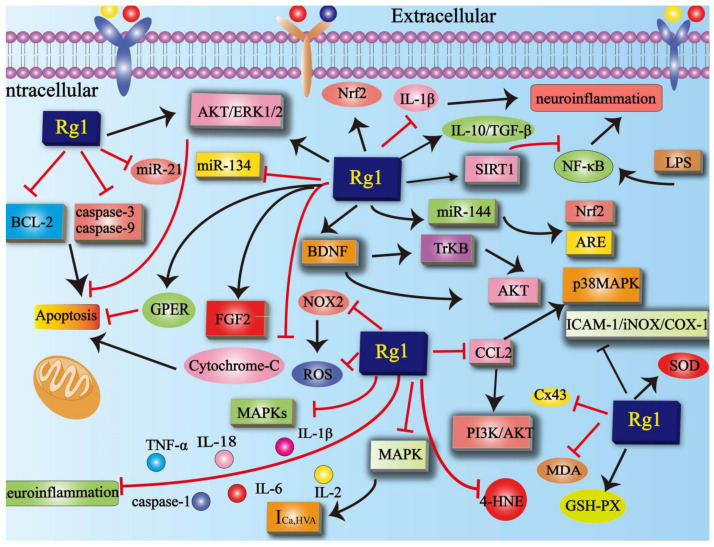
Mechanisms of Rg1 in nervous system diseases. Rg1 inhibits apoptosis and reduces neuroinflammatory response in neurological diseases. Rg1 inhibits the expression of the MAPK (ERK1/2, INK, and p38 MAPK) family in neurological diseases. Rg1 improves oxidative stress in neurological diseases by reducing the level of MDA, ROS, NOX2, and 4-HNE while increasing the level of SOD and GSH-px. SOD = superoxide dismutase; GSH-pX = glutathione peroxidase; MDA = malondialdehyde; NOX2 = NADPH oxidase 2; 4-HNE = 4-hydroxytryptamine; ROS = reactive oxygen species; MAPK = mitogen-activated protein kinase; CCL2 = C-C motif chemokine-2; I_Ca,HVA_ = high-voltage-activated calcium currents; PI3K = phosphatidylinositol 3-kinase; Akt = protein kinase B; Nrf2 = nuclear factor-erythroid 2-related factor 2; ARE = antioxidant response element; miR-144 = microRNA 144; miR-873-5p = microRNA-873-5p; miR-134 = microRNA-134; NF-κB = nuclear factor-κB; SIRT1 = sirtuin-1; TNF-α = tumor necrosis factor-α; IL = interleukin; miR-21 = microRNA-21; BDNF = neurotrophic factor; Cx43 = Connexin43; ICAM-1 = intracellular adhesion molecule 1; COX2 = cyclooxygenase-2; iNOS = inducible nitric oxide synthase; FGF2 = fibroblast growth factor 2; GPER = G-protein-coupled estrogen receptor 1; TrkB = Tyrosine Kinase receptor B; Bcl-2 = B-cell lymphoma-2.

**Table 1 cells-11-02529-t001:** The summary of Rg1 molecular mechanisms in neurological diseases.

Model/Disease	Mechanisms	Reference
CUMS	Inhibited the increase in the expression levels of caspase-3 and caspase-9	[44]
	Decreased the expression level of Bcl-2	[44]
	Inhibited the expression of Nrf2 and inhibited the activation of p38 mitogen—activated protein kinase (p-p38 MAPK) and p65 subunit	[44]
AD	Inhibited neuronal apoptosis by regulating the expression of miR-873-5p	[57]
Aging	Restored FGF2-Akt and BDNF-TrkB signaling pathways to inhibit neuronal apoptosis	[58]
Neurological symptoms and cauda equina syndrome	Downregulated caspase-3 expression	[58]
Induced by Aβ25-35	Inhibited apoptosis through Akt and ERK signaling	[59]
PD	Reduced the production of ROS and the release of mitochondrial cytochrome-C into the cytoplasm, and subsequently inhibited the activation of caspase-3	[60]
Depression	Inhibited hippocampal neuronal apoptosis by GPER	[61]
CSDS	Inhibited the release of IL-6, IL-1β, and TNF-α, as well as NF-κB via the MAPK and SIRT1 signaling pathways.	[63]
PD	Increased anti-inflammatory cytokines including TGF-β, IL-10, and BDNF secretion to protect neurons	[64]
Behavioral deficits	Reduced the level of ICAM-1, COX-2 and iNOS as well as maintained the integrity of BBB permeability	[65]
Depression	Reduced the levels of IL-1β, TNF-α, caspase-1, IL-2, IL-6 and IL-18 via suppression of Cx43 ubiquitination in depression	[66]
Ischemic/reperfusion	Protected neuronal injury by regulating miR-144, which regulated Nrf2/ARE signaling via miR-144	[67]
Chronic stress-induced	Blocked the function of miR-134 and significantly improved neuronal structural abnormalities, biochemical changes and depression-like behavior	[68]
Aβ-exposed	Inhibited the expression of high-voltage-activated calcium currents (ICa,HVA) in hippocampal neurons of Aβ-exposed rat brain slices through MAPK	[105]
Depression	Attenuated Ly6Chi monocyte pro-inflammatory factor release in the blood	[69]
Depression	Enhanced the activity of SOD and GSH-pX while reducing the level of ROS and MDA in the hippocampal CA1 region	[35]
H_2_O_2_-treated hippocampal neurons	Reduced the level of NOX2	[71]
Dopamine-induced PC12 cells	Ameliorated apoptosis by inhibiting oxidative stress	[17]
Depression	Downregulated 4-HNE in the CA1 region of the hippocampus	[71]
Cognitive Impairment	Reduced the level of ROS and increased the activities of SOD and GSH-pX	[72]
Cadmium-induced neurotoxicity	Attenuated oxidative stress and inflammation	[74]
AD	Improved oxidative stress, apoptosis, and neuroinflammation	[75]
PD	Mediated the differentiation of microglia into the M2 phenotype via the NF-κB pathway	[64]
HD	Prevented the death of striatal neurons by inhibiting the activation of MAPKs and the NF-κB pathway	[84]
AD	Regulated the protein expression and improved memory impairment	[87]
SAMP8 mice	Ameliorated the escape acquisition and reversed memory deficits	[90]
CRS	Improved learning and memory	[91]

## Data Availability

Not applicable.

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
