# Peer review of "New Therapeutic Approaches to and Mechanisms of Ginsenoside Rg1 against Neurological Diseases"

_cells, 2022, doi:10.3390/cells11162529_

Round 1

Reviewer 1 Report

In this second submission the authors have extensively modified the body of the text, significantly improving its quality. Below are the comments to the authors for further changes to the text before a possible publication on this journal

-In figure 4 the role of miR-21 is not clear; the authors can briefly indicate this information in the caption as they did for the other molecules.

-A spell check of the English form by a native English speaker is recommended.

Author Response

Thank you very much for your comments. We have indicated the mechanism of miR-21 in figure 4 caption. And I have asked the English editor for help again in MDPI. The advice is that we can manuscript it if we are ready. Then we checked it again.

Reviewer 2 Report

The manuscript has been substantially improved

Author Response

Thank you for reviewing the manuscript again. At the same time, thank you for your comments and suggestions when reviewing the manuscript for the first time, so that our manuscript can be improved.

Reviewer 3 Report

This is much improved, but there are still some problems with English. For example,"Overall, Rg1 involves fewer disease models by inhibiting the neuroinflammatory response, and it is more about inhibiting the expression of inflammatory factors." What does involve a disease model mean? Or "Rg1 improves neurological diseases by regulating the MAPK family.." and "Therefore, Rg1 can improve oxidative stress by acting on neuronal cells in vitro. ". Do you mean control it or increase it? Another example is "In this review, we focused on the therapeutic potential of Rg1 for improving neurological diseases in this review." So that is repetitive. There are not many problems, but a review of the writing is needed.

Author Response

Thank you very much for your comments. We have reviewed the manuscript according to your requirement. We changed “ Rg1 involves fewer disease models by inhibiting the neuroinflammatory response” to “there are few disease model studies on Rg1 by inhibiting the neuroinflammatory response”. We have changed it to: Rg1 improves neurological diseases by inhibiting the expression of MAPK family molecules. We have changed it to: Rg1 can inhibit oxidative stress by acting on neuronal cells in vitro.We deleted “ in the review ” in line 27.

This manuscript is a resubmission of an earlier submission. The following is a list of the peer review reports and author responses from that submission.

Round 1

Reviewer 1 Report

This review provides a valuable insight into the molecular mechanisms that underlie the therapeutic potential of Ginsenoside Rg1 in the context of neurological diseases. This is a well compiled review encompassing contemporary studies. The following suggestions may increase the merit of the manuscript. 

1. The title of the manuscript could be rephrased such that it better embodies the essence of the review.  

2. The Abstract could be rewritten and rephrased to better represent and reflect the contents of the review. 

3. The Figure legends could be expanded to provide a better delineation of the illustrated figures depicting the effects of Ginsenoside Rg1 on the potential signaling mechanisms involved. 

4. The General Discussion section (Section 7) could be expanded to provide a greater insight into the individual facets and topics encompassed by the review. 

5. The authors could add a "Conclusion" section to reiterate and provide an integrated outlook of the therapeutic potential of Ginsenoside Rg1. 

Reviewer 2 Report

The authors in this review say they provide a theoretical basis for the in-depth study of Rg1 and provide new insight and ideas for the clinical treatment of nervous system diseases. The review, although deserving as a discussion, cannot be published in this form. The review did not achieve the objectives mentioned and presents a rather superficial and partial treatment on the subject. Below are the comments to the authors.

Abstract section

-          It is not correct what the authors say regarding the total absence of revision on the role of RG1 in the nervous system, as in the literature there are also verifiable contributions on PubMed. The authors can explain that the purpose of this review is to dig deeper ideas for the clinical treatment of nervous system related diseases, specifying what diseases of the nervous system. The abstract should also capture the attention and interest in reading the review, so the authors must better specify the conceptual meaning of Rg1, because for many readers this acronym may not mean anything.

Manuscript Body

-          In general terms, authors must make style as well as grammatical corrections on the entire text of the manuscript. A careful linguistic revision of English is advisable, because the grammatical style leaves much to be desired

-          5.2. Effects of Rg1 on neuroinflammation: Through which mechanisms does Rg1 exert its anti-inflammatory role? in addition to citing the effects, it is important in which models these effects have been described and through which molecular mechanisms Rg1 acts.

-          Line 159: please correct MIRNA

-          Line 163 : in which animal model?

-          Line 197 and Figure 6: In the figure 6 it is not clear that Rg1 increases SOD as the authors report in the text, but it appears that it is blocked.

-          5.8. Effects of Rg1 on gut microbiota: I believe that this sentence is completely insufficient to justify the existence of a specific paragraph dedicated to the role of Rg1 on the gut microbiota. As this is a review, the authors should be very thorough on these topics.

Conclusion

-The problem of pharmacokinetics and bioavailability is a problem that does not exclusively concern Rg1, but several bioactive compounds. An example that has not been addressed in this review for example is the information on the ability of Rg1 to cross the blood brain barrier or seal its integrity.

In general, however, in this review presented as a work containing information on the role of Rg1 in the nervous system, it does not give the impression of being very comprehensive. Perhaps it could be presented as a mini-review, since, apart from the numerous figures, in the individual sections the topics are just mentioned without providing a real insight into either the molecular mechanisms or the experimental models in which some observations were conducted.

Reviewer 3 Report

THE PARAGRAPH "4. Neurological diseases" should be amplied and described at the beginning of the review, underlying the different molecular mechanisms involved in these pathologies. 

- The figures are well structured. Nevertheless, the authors should better describe the figure captions. 

-a table summarising Rg1 molecular targets would be useful

Reviewer 4 Report

The English writing is very poor. There are many incorrect sentences, punctuation and capitalization. There are even some issues with font types. Also, some sections repeat themselves, such as listing types of PPT and PPD ginsenosides. Incorrect words are used in many places, such as "apologize" in the acknowledgments. The figure legends need to be more descriptive with definitions for the abbreviations. Most sections simply list results of studies without any attempt at integration, comparisons, analysis or conclusions. So it reads mostly like a list of results. The discussion section should be giving conclusions based on the previous sections, rather than introducing new material. For example, the previous sections should mention about clinical studies for each aspect, so that a conclusion about can be made. Overall, this needs very extensive rewriting.